# A general method for estimating the prevalence of influenza-like-symptoms with Wikipedia data

**Giovanni De Toni**[1]*, **Cristian Consonni**[2], **Alberto Montresor**[1]

**1** Department of Information Engineering and Computer Science (DISI), University of Trento, Trento, Italy, **2** Big Data and Data Science Unit, Eurecat - Centre Tecnològic de Catalunya, Barcelona, Spain

* giovanni.detoni@unitn.it

**Data Availability Statement:** Wikipedia and Influenza Dataset: Zenodo, doi:10.5281/zenodo.2248500 Models and code: Github, https://github.com/fluTN/influenza The dataset and code's repository are also reported inside the manuscript.

## Abstract

Influenza is an acute respiratory seasonal disease that affects millions of people worldwide and causes thousands of deaths in Europe alone. Estimating in a fast and reliable way the impact of an illness on a given country is essential to plan and organize effective counter-measures, which is now possible by leveraging unconventional data sources like web searches and visits. In this study, we show the feasibility of exploiting machine learning models and information about Wikipedia's page views of a selected group of articles to obtain accurate estimates of influenza-like illnesses incidence in four European countries: Italy, Germany, Belgium, and the Netherlands. We propose a novel language-agnostic method, based on two algorithms, *Personalized PageRank* and *CycleRank*, to automatically select the most relevant Wikipedia pages to be monitored without the need for expert supervision. We then show how our model can reach state-of-the-art results by comparing it with previous solutions.

## 1 Introduction

Influenza is a widespread acute respiratory infection that occurs with seasonal frequency, mainly during the autumn-winter period. The European Centre for Disease and Control (ECDC) estimates that influenza causes up to 50 million cases and up to 70, 000 deaths annually in Europe [1, 2]. Globally, the number of deaths caused by this infection ranges from 290, 000 to 650, 000 [3]. The most affected categories are children in the pediatric age range and seniors. Severe cases are recorded in people above 65 years old and with previous health conditions, for example, respiratory or cardiovascular illnesses [4, 5].

The ECDC monitors the influenza virus activities and provides weekly bulletins that aggregate the open data coming from all the European countries. Local data are gathered from national networks of volunteer physicians [6]. The Centre supplies only data about the current situation, which means that they do not provide information about the evolution of the spread of influenza, namely a forecast on the number of infected people. Moreover, ECDC data are available with a delay ranging between one and two weeks, preventing taking precautionary measures to mitigate its impact. Estimating influenza-like illnesses diffusion in a population is

**Funding:** The author(s) received no specific funding for this work.

**Competing interests:** The authors have declared that no competing interests exist.

a highly challenging problem whose solution can provide many benefits. The need for finding such a solution prompted the creation of national challenges to discover better methods for rapid nowcasting and forecasting of influenza [7].

In recent years, thanks to machine learning techniques, new methods have been developed to estimate the activity level of diseases by harnessing new data sources made available from the usage of internet services and big data. Researchers have tried for many years using unconventional sources of data to make predictions, for example, tweets [8–10], Google's search keywords [11, 12], Google Flu Trends [13] social media [14], Internet-based surveys [15], and mobility and telephony data [16–18]. Other works have employed multiple data sources to improve prediction and forecasting capabilities [19]. A variety of methods have been applied, from purely statistical approaches, autoregressive models, linear models and generalized linear models to neural networks [12, 20]. Interestingly, almost all machine learning approaches described in the literature to date use a fixed set of features chosen manually by human experts.

Wikipedia [21] is a multilingual encyclopedia written collaboratively by volunteers online. As of this writing, the English version alone has more than 6.2M articles and 50M pages, and it is edited on average by more than 65k active users every month [22].

Research has shown that Wikipedia is a first-stop source for information of all kinds, including science [23, 24]. This is of particular significance in fields such as medicine, where it has been shown that Wikipedia is a prominent result for over 70% of search results for health-related keywords [25]. Wikipedia is also the most viewed medical resource globally [26], and it is used as an information source by 50% to 70% of practising physicians [27], and medical students [28, 29].

In this study, we investigate how to exploit Wikipedia as a source of data for rapid nowcasting of influenza activity levels in several European countries. In 2014, McIver and Brownstein [30] developed a method based on Wikipedia's pageviews, which behaves quite well when predicting influenza levels in the United States provided by the Center for Disease Control and Prevention (CDC). They used pageviews of a predefined set of pages from the English language edition of Wikipedia chosen by a panel of experts. The same year, Generous et al. [31] developed linear machine learning models and demonstrated the ability to forecast incidence levels for 14 location-disease combinations.

Selecting the right pages to feed into these models is additionally challenging, Priedhorsky et al. [32] showed the difficulty of selecting good predictors from Wikipedia. The model can happen to choose pages that correlate with influenza but that are unrelated and potentially misleading. In the same work, it is shown how adding confounder pages, such as *"Basketball"* or *"College basketball"*, can reduce the nowcasting and forecasting performances. These results also underline the importance of choosing the proper technique to select a suitable set of Wikipedia pages to monitor, including fewer confounders as possible.

In this work we address this challenge and we extend upon the state-of-the-art by proposing a novel method that can be applied automatically to other languages and countries. We focused our attention on predicting influenza levels in Italy, Germany, Belgium and the Netherlands. We have developed a method that is more flexible with respect to the characteristics of the edition of Wikipedia, and that can be applied to any language, thus enabling us to build and deploy machine learning models for *Influenza-like Illnesses (ILI)* estimation without the need of expert supervision.

The paper is structured as follows: in Section 2, we review the relevant datasets that we have used in our study. In Section 3 and 4, we discuss our models and how we selected the relevant features to use for predicting influenza activity levels from Wikipedia data. We present the

results of our study in Section 5 and 6, and we discuss them in Section 7. Finally, we conclude the paper in Section 8.

## 2 Data sources

### 2.1 Wikipedia page views

Data about Wikipedia page views are freely available to download. For our study, we have used two datasets: the `pagecounts` dataset [33], available from December 2007 to May 2016, for a total of 461 weeks, and the `pageviews` dataset, available from October 2016 to April 2019, for a total of 134 weeks. Both of them count the number of page hits to Wikipedia articles. The `pagecounts` dataset contains non-filtered page views, including automatically bot-generated ones; moreover, it does not contain data of page hits from mobile devices. The `pageviews` dataset has been developed more recently, superseding `pagecounts` in October 2016, and includes only human traffic from both desktop and mobile devices. By considering both these datasets, we can examine a more extended time range while estimating the impact of different models on our results.

We extracted the page views of three different versions of Wikipedia from these datasets: Italian, German, and Dutch. We selected these languages because for each of them, most page views come almost exclusively from one single area [34]. For instance, just 41.5% of the page views of the English Wikipedia come from the US and the UK. On the contrary, 87.0% of the page views of the Italian version of Wikipedia come from Italy; 77.0% of the page views of the German version of Wikipedia come from Germany, and 8.0% come from Austria; finally, almost 89% of the Dutch Wikipedia's page views come from the Netherlands (69.1%) and Belgium (20.3%).

The provenance of visits is measured by looking at the percentage of aggregated unique IP addresses coming from the given country, while the IPs of requests made by bots are not included.

The Wikimedia Foundation, Inc. –the non-profit organization which runs Wikipedia's servers –provides information about countries only at an aggregated level; in fact, the IP address of individual requests is very sensitive information that, if exposed, could jeopardize the privacy and safety of Wikipedia readers.

In both the `pagecounts` and `pageviews` datasets, the following data are available: the project's language, the page title, the number of requests and the size in byte of the page retrieved. Data are available with hourly granularity. For our study, we filtered them to select only the entries related to given sets of encyclopedic articles, selected as described below. Then we computed aggregated weekly values of the page views for those articles.

The period we considered in our study started on the 50th week of 2007 and ends on the 15th of 2019, for a total of 591 weeks. `pageviews` data were used for September 2016-April 2019, while for the previous period `pagecounts` data were used.

We preprocessed the datasets to address some issues related to the availability and standardization of data before feeding them to our models. First, since Wikipedia articles are created collaboratively by the community of editors, their creation date may vary: hence page views may be missing for a certain period for the simple reason that an article did not exist at the time. Second, recently-created Wikipedia pages may be moved, i. e. their title can be changed, they are also less well-connected to existing Wikipedia articles, which influences the number of incoming readers to those pages. Therefore, we removed those pages which had more than 50% of missing page views values. We also assigned the value of zero to the remaining missing page views. Third, we performed a Yeo-Johnson transformation [35] on Wikipedia's data to improve heteroscedasticity and to ensure normality. Finally, we standardized

them by removing the mean and then dividing them by the standard deviation. This standardization has been performed on the training set before training the model; however, for the test set, we used the same mean and standard deviation calculated from the training set to avoid introducing a bias in the test data. The training data are the same that we assume would be available in an actual nowcasting scenario. We included the week number as a feature of our model. In order to do so, we one-hot encoded the week value into binary vectors $w_i \in \{0, 1\}^{52}$ where the bit corresponding to the given week is set to 1.

We then built three different datasets named: PC, PV and PC+PV. PC contains exclusively the pagecounts data, therefore covering eight influenza seasons (from 2007-2008 to 2014-2015). PV contains only the pageviews data, thus covering only four influenza seasons (from 2015-2016 to 2018-2019). PC+PV was obtained by merging pagecounts and pageviews data, thus covering all the influenza seasons (from 2007-2008 to 2018-2019).

## 2.2 Influenza activity levels

Data about the incidence of influenza in the population are available through different online official websites and repositories. We collected influenza data for these countries: Belgium, Italy, Germany, and the Netherlands. In the following, we briefly describe the collection procedure we followed for each country.

Italian data were downloaded from the InfluNet system [36], which is the Italian flu surveillance program managed by the Istituto Superiore di Sanità; German data were obtained from the Robert Koch Institute; [37] Dutch and Belgian data were taken from the GISRS (Global Influenza Surveillance and Response System) online tool which is available on the WHO website [38], at the best of our knowledge, the two countries do not offer open data through their national health institute websites.

For Germany and Italy, we cover 12 influenza seasons, from 2007-2008 to 2018-2019; for the Netherlands and Belgium, we are limited to 9 influenza seasons, from 2010-2011 to 2018-2019. Each of the datasets records the influenza incidence aggregated weekly over 100000 patients. For each influenza season, each entry represents the influenza level for a specific week.

Each influenza season consists of 26 weeks, from the 42nd week of the previous year to the 15th week of the following year; for instance, the 2009-2010 influenza season comprises data ranging from the 42nd week of 2009—starting on Monday, October 12th—to the 15th week of 2010—ending on Sunday, April 15th. The complete dataset contains 312 weeks for Germany and Italy, and 234 for the Netherlands and Belgium. No preprocessing was applied to the incidence data.

## 3 Feature selection

During the influenza seasons, we should see an increment in the page view number of some specific Wikipedia's articles, which should be related to flu topics (e.g., *"Influenza"*, *"Headache"*, *"Flu vaccine"*, etc.). That would occur because most people tend to search information online about their symptoms and their disease when they are sick, leading to an increase in the page views for those articles.

However, we do not choose which pages to feed into the model manually. Instead, we use three different methodologies to avoid making further unnecessary assumptions.

### 3.1 Wikipedia's categories

The first method consists of determining a series of Wikipedia's categories related to the medical sector to extract lists of articles. We started by choosing the categories from the Italian

**Table 1. Selected Wikipedia categories.** We first selected the Italian categories, and then we chose the corresponding German and Dutch translations. We report the English categories for reference.

| English | Italian | German | Dutch |
|---|---|---|---|
| Viral diseases | Malattie infettive virali | Virusziekte | Virale Infektionskrankheit |
| Infectious diseases | Malattie infettive | Infectieziekte | Infektionskrankheit |
| Epidemics | Epidemie | Epidemie | Epidemie |
| Viruses | Virus | Virus | Viren, Viroide und Prionen |
| Vaccines | Vaccini | Vaccin | Impfstoff |
| Medical signs | Segni clinici | Symptoom | Krankheitssymptom |

version of Wikipedia. After doing that, we replicated the procedure for all the other languages (German, Dutch) by selecting the corresponding translation of the given category in the Italian Wikipedia. We ended up having a dedicated set of Wikipedia pages for each of the different Wikipedia versions. In the end, we produced three different lists of Wikipedia pages, which were used to filter the Wikipedia page view dumps.

The categories selected by this method can be seen in Table 1.

## 3.2 Graph-based methods

The strategies that we propose in this section can choose automatically the set of relevant pages to monitor. In this way, we could keep our page list always up-to-date with minimum effort yet retaining its effectiveness.

We can represent Wikipedia as a graph in which each article is a node, and links between them are directed edges. Wikipedia editors write articles and insert links to other pages which are contextually relevant; readers also use links to navigate between pages. When choosing which pages are relevant, we should consider the entire graph structure; for example, we may consider the pages linked by the *"Influenza"* article to be relevant, but also the ones that link to it. Previous work has shown that Wikipedia links provide a way to explore the context of an article [39] and we can exploit this knowledge to extract the best pages to monitor. Ideally, we would like to be able to give each Wikipedia page a score that measures its relative importance to a given "source" page, in this case, the *"Influenza"* page for each language.

Several algorithms can be used to given scores to graph nodes based on specific metrics. We adopt here two approaches: the well-known Personalized PageRank [40] and a more recent algorithm, called CycleRank [41].

**3.2.1 Personalized PageRank.** The Personalized PageRank algorithm [40] (*PPageRank* for short) ranks web pages in order of importance with respect to a set of *source pages*. *PPageRank* models the relevance of nodes around the selected nodes as the probability of reaching each of them when following random walks starting from one of the sources nodes.

As this algorithm has been applied successfully to a broad range of graph structures in order to find relevant items [42, 43], we include it as a baseline in our analysis. Its performance, however, is hindered by pages with high in-degree that function as hubs and obtain high scores regardless of the starting point.

**3.2.2 CycleRank.** The *CycleRank* algorithm [41] is a novel approach based on cycles, i.e. circular walks that start and end in a given node, exploiting both incoming and outgoing links to reduce the importance of in-degree hubs. The idea behind this method is that by using circular walks, it is possible to identify pages that are more pertaining to the context of the topic of interest. Cycles guarantee that we rank higher pages that are, at least indirectly, both linked from and linking to the reference article, thus avoiding the pitfalls of *PPageRank* where

random walks may easily lead far away from the original topic, into pages that are not relevant at all.

The *CycleRank* score can be seen as proportional to the time spent on a given node when following random loops from the reference node. Similarly to *PPageRank*, *CycleRank* can be interpreted as the weighted probability of ending up in a node when following a random path that starts and ends in the source nodes. Furthermore, *CycleRank* gives a non-zero scores only to nodes that are reachable from the reference node through circular paths, thus automatically filtering a subset of pages connected to the *"Influenza"* page.

### 3.3 Discussion

We then analyze the set of features extracted by using these different methodologies. For the Dutch, German and Italian Wikipedia versions, respectively, the *Categories* feature sets contain 382, 381 and 470 Wikipedia's pages; the *PPageRank* feature sets contain 274, 277 and 245 pages; and the *CycleRank* feature sets contain 86, 237 and 160 pages.

The feature set built from the categories contains more pages than the other two. We expect this behaviour since we do not impose any constraint on them. The *CycleRank* and *PPageRank* feature sets are smaller since we enforce that all those pages must be connected to the *"Influenza"* page; *CycleRank* sets are the smallest ones because they require connections in both directions. In general, the *Categories* feature set encompasses a broad spectrum of pages related to medical topics, which may or may not have any relation with the *"Influenza"* concept.

Table 2 shows how much the features sets overlap across the three methods under consideration. *CycleRank* and *PPageRank* feature lists share many common pages. For instance, 42% of the pages selected by *CycleRank* are also selected by the *PPageRank*, while *Categories* feature sets are more diverse, with many pages appearing only with this method. However, for the Dutch Wikipedia, *CycleRank* shares 42% of its features with the *Categories* Dutch feature set.

At first glance, the *Categories* method outlined above is complex and prone to errors. Choosing the correct pages to monitor is a time-consuming task that requires some expertise in the medical and epidemiological domains. Moreover, creating and maintaining this list of pages is a process that needs to be done manually and frequently. Wikipedia's structure can undergo rapid changes, and each of its pages can be deleted, moved or renamed, thus forcing us to keep our list of features constantly updated. On the other hand, *CycleRank* and *PPageRank* do not require human intervention, and they are indeed faster than the manual *Categories*.

### 4 Models

We trained two different models: a simple linear model regularized with the Least Absolute Shrinkage and Selection method (LASSO) [44] and a Poisson Generalized Linear Model (GLM) regularized through elastic net [45]. The regularization diminishes the possible overfitting and automatically reduces the number of features used by the models. The models were

**Table 2. Percentages of features shared by each feature set given the method used to extract them.** Since the various datasets have different sizes, the percentages are not symmetrical. Each row shows the result for one of the methods. For instance, the *CycleRank* row shows the fraction of features it shares with the other feature sets.

| | CycleRank | | | PPageRank | | | Categories | | |
|---|---|---|---|---|---|---|---|---|---|
| | *Italian* | *German* | *Dutch* | *Italian* | *German* | *Dutch* | *Italian* | *German* | *Dutch* |
| **CycleRank** | 100% | 100% | 100% | 53,8% | 49,4% | 83,7% | 18,8% | 11,0% | 41,9% |
| **PPageRank** | 35,1% | 42,2% | 26,3% | 100% | 100% | 100% | 10,2% | 8,3% | 15,3% |
| **Categories** | 6,4% | 6,8% | 9,4% | 5,3% | 6,0% | 11,0% | 100% | 100% | 100% |

trained by stochastic gradient descent (SGD) and by trying to minimize the Mean Squared Error (MSE). During training, we used cross-validation to estimate the perfect balance for the regularization terms (e.g., $\lambda$ parameter of $\min_w \frac{1}{2n} \| y - Xw \|_2^2 + \lambda R(w)$ where $R(w)$ is the regularizer). We decided to use a simple linear model to provide a baseline and then a more sophisticated GLM, which is already widely used in the literature [12, 13, 30]. Moreover, we can exploit the model's weights to infer each page's relative importance towards the final prediction.

We tested six different models. Three with LASSO: *Categories*, *PPageRank* and *CycleRank*. Three with Poisson GLM: *Categories-GLM*, *PPageRank-GLM* and *CycleRank-GLM*. Each of them reflects the set of features used. For simplicity, later in the text, we might omit the *-GLM* marking, but it will be clear from the context. Given a model and target influenza season, we train the model by keeping as the test set only the target season data. The training set is formed by the remaining influenza seasons. While training, we use 5-fold cross-validation to estimate the optimal regularization weight $\lambda$. By keeping fixed the model, this procedure is repeated for each influenza season. Ultimately, we compute the evaluation metrics for each influenza season separately, and then we average the outcomes to provide the final result. Apart from the last season, every model has been evaluated on a given influenza season (e.g., 2015-2016), but the model has been trained by also using future influenza season data (e.g., 2016-2017, 2017-2018, etc.). We argue that this is acceptable since we assume the various influenza period to be conditionally independent. Even though many factors could make this assumption less precise, we claim that this experimental setting is still appropriate to give us an idea about the performance of the final complete models, which will be used to estimate the ILI case on new data. The only exception to this is the PC dataset. In that case, we trained the models by only using data from previous seasons (e.g., from 2007-2008 to 2014-2015), and therefore future seasons are not included in the training set.

The code (https://github.com/fluTN/influenza) and the dataset [46] used for the project are released as an open-source project to ensure proper peer-review and to assure reproducibility of our results. We implemented the LASSO models by using the scikit-learn library [47], while for the GLM models we used statsmodels [48] and Pyglmnet [49] libraries.

## 5 Results

We compared the performance of the *CycleRank* and *PPageRank* models against the one trained with the user-defined categories (*Categories*). We trained the models to predict the last four influenza seasons (2015-2016, 2016-2017, 2017-2018 and 2018-2019). We performed this analysis for all the chosen European countries: Italy (IT), Germany (DE), Belgium (BE) and the Netherlands (NL). We measured the Pearson Correlation Coefficient (PCC) and the $R^2$ score.

We also evaluated the accuracy in estimating the week in which the influenza peak occurs (highest incidence value over the entire season). We considered a peak estimation acceptable if the model was able to estimate the peak with a deviation of $\pm 2$ weeks with respect to the ground truth. The models were trained by using the PC, PV and PC+PV datasets. Namely, we wanted to analyze the effects of providing additional data during training, even if it was recorded with a different methodology. In this work, we also tried to evaluate the goodness of the most critical features selected by the various models. We tried to understand which Wikipedia's pages are the most valuable when estimating the influenza incidence over a population. Moreover, we also wanted to assess the effectiveness of our automatic method against the traditional, manual one (*Categories*).

## 5.1 Estimation accuracy

Table 3 shows the Pearson Correlation Coefficient for each model and each country for both LASSO and GLM.

Since models trained with the PC dataset perform worse than models trained with either PV or PV+PC, we did not perform additional analysis of the former. Moreover, Wikipedia discontinued such a dataset in favour of PV. Therefore, models trained with PC may become obsolete in the future because of distribution shift or other phenomena. Interestingly, there are some cases in which models trained with the PC datasets outperform PV+PC (IT *CycleRank-GLM*, IT *PageRank-GLM* and BE *Categories*). This might happen because the PC dataset adds additional noise or bias to the PV data, making the final model converge to a suboptimal configuration.

We focused instead on PV and PV+PC for the rest of the analysis. We present a comparison between the performances of PV+PC and PV.

All models present a high PCC with the influenza incidence data ($p < 0.005$). Generally, *CycleRank* and *PPageRank* yield better results or provide similar performances to the *Categories* model. For the LASSO models, the *PPageRank* model scores the highest correlation coefficient on 3 out of 4 examined countries (IT, DE, BE), with the best model for NL being *CycleRank*. For the GLM models, the *CycleRank* model scores the highest PCC in all four examined countries.

By examining the effect of using different datasets, in 9 over 12 experiments for the LASSO models, the PC+PV dataset improves the PCC by about 7.3% on average. In one experiment, using either dataset produce the same performance. This effect is visible for *CycleRank* and *PPageRank*, while *Categories* seem not to obtain the same gain when employing a larger page view dataset. In the remaining three experiments, we noticed that using a more extensive training dataset produced an average PCC loss of around 2.4%.

For the GLM models, in 8 over 12 experiments, the PC+PV dataset improves the PCC by about 7% on average. In the four other cases, the average PCC loss is around 4.5%.

We also compare the best LASSO models against the best GLM models. Table 4 shows the results. Clearly, GLMs outperform LASSO models in 9 out of 12 experiments. The GLM improves over LASSO of about 7.4% on average. Table 5 shows the $R^2$ score for the best LASSO and GLM models for each country. We can notice how they show high positive values, and again the GLM models outperform LASSO in 9 experiments out of 12.

**Table 3. Mean Pearson Correlation Coefficient.** The bold values indicate the best results over all combinations of models and datasets. The green colour indicates the best results by either using the PC, the PV or the PC+PV dataset, by keeping both the country and the model fixed (no colour means no changes).

| Country | CycleRank | | | PageRank | | | Categories | | |
|---|---|---|---|---|---|---|---|---|---|
| | PC | PV | PV+PC | PC | PV | PV+PC | PC | PV | PV+PC |
| IT | 0.875 | 0.918 | 0.918 | 0.870 | 0.903 | **0.933** | 0.823 | 0.869 | 0.896 |
| DE | 0.697 | 0.675 | 0.844 | 0.799 | 0.671 | **0.874** | 0.435 | 0.614 | 0.597 |
| NL | 0.732 | 0.859 | **0.871** | 0.786 | 0.824 | 0.854 | 0.770 | 0.754 | 0.814 |
| BE | 0.737 | 0.720 | 0.806 | 0.826 | 0.793 | **0.836** | 0.736 | 0.763 | 0.734 |

| Country | CycleRank-GLM | | | PageRank-GLM | | | Categories-GLM | | |
|---|---|---|---|---|---|---|---|---|---|
| | PC | PV | PV+PC | PC | PV | PV+PC | PC | PV | PV+PC |
| IT | 0.897 | **0.928** | 0.849 | 0.865 | 0.881 | 0.837 | 0.813 | 0.869 | 0.840 |
| DE | 0.931 | 0.879 | **0.965** | 0.931 | 0.903 | 0.948 | 0.580 | 0.680 | 0.763 |
| NL | 0.867 | **0.915** | 0.887 | 0.830 | 0.793 | 0.866 | 0.719 | 0.746 | 0.844 |
| BE | 0.873 | 0.791 | **0.891** | 0.844 | 0.837 | 0.884 | 0.501 | 0.730 | 0.756 |

**Table 4. PCC comparison between LASSO and GLM best models.** The bold values indicate the best results. The green colour indicates the best results by either using the `GLM` or the `LASSO` model, by keeping the country fixed (no colour means no changes).

| Country | CycleRank | | PageRank | | Categories | |
|---------|-----------|-----|----------|-----|------------|-----|
| | **LASSO** | **GLM** | **LASSO** | **GLM** | **LASSO** | **GLM** |
| IT | 0.918 | 0.928 | **0.933** | 0.881 | 0.896 | 0.869 |
| DE | 0.844 | **0.965** | 0.874 | 0.948 | 0.614 | 0.763 |
| NL | 0.871 | **0.915** | 0.854 | 0.866 | 0.814 | 0.844 |
| BE | 0.806 | **0.891** | 0.836 | 0.884 | 0.763 | 0.756 |

**Table 5. $R^2$ comparison between LASSO and GLM best models.** The bold values indicate the best results. The green colour indicates the best results by either using the `GLM` or the `LASSO` model, by keeping the country fixed (no colour means no changes).

| Country | CycleRank | | PageRank | | Categories | |
|---------|-----------|-----|----------|-----|------------|-----|
| | **LASSO** | **GLM** | **LASSO** | **GLM** | **LASSO** | **GLM** |
| IT | 0.733 | 0.649 | **0.803** | 0.645 | 0.519 | 0.608 |
| DE | 0.408 | **0.916** | 0.418 | 0.839 | 0.088 | 0.392 |
| NL | 0.508 | **0.746** | 0.455 | 0.624 | 0.300 | 0.538 |
| BE | 0.347 | 0.399 | 0.517 | **0.641** | 0.434 | 0.414 |

Figs 1 and 2 show the predicted influenza trend for each of the given countries when using the best LASSO and GLM models, respectively. Sometimes the models produce visible "spikes". Those outliers are related to a sudden page views increment (or decrement) of one or more feature pages. Many factors could cause these sudden changes in page views counts. For instance, unexpected news coverage of a specific topic may artificially increase the total views of a particular page, thus leading to overestimation of the results.

Ideally, we want our model to be robust against these kinds of situations, and we prefer models that generate fewer "spikes". This objective also reflects on the set of pages we are using to monitor ILI incidence. We want our model to select highly informative pages that do not present sudden changes in the page views count. The graphs qualitatively show how the GLM models provide better estimates for influenza incidence. For instance, they are able to capture the 2016-2017 season correctly both for Germany and the Netherlands, while LASSO models fail to do so. Generally, we can see from the graphs how the models trained with the *Categories* feature set (green lines) are not able to capture certain influenza seasons. On the contrary, by using *CycleRank* and *PPageRank* feature sets, we have models which are more resilient and produce better predictions.

## 5.2 Peak accuracy

Table 6 shows the peak prediction performance for LASSO and GLM models, respectively. The models are able to estimate the highest influenza peak within a 2-week range with respect to the ground-truth value. Unfortunately, they do not reach a level of accuracy high enough to estimate precisely the peak. By examining the LASSO models trained with the extended dataset (`PC+PV`), in 8 models over 12 we can see a small increment in the correctly predicted peaks. On the GLM models, the improvement is modest. Only in 5 models over 12 we can see an improvement.

## 6 Feature analysis

In this section, we analyze which features are selected by each model to characterize how the chosen features affect the final estimate. Namely, which are the best predictors for estimating

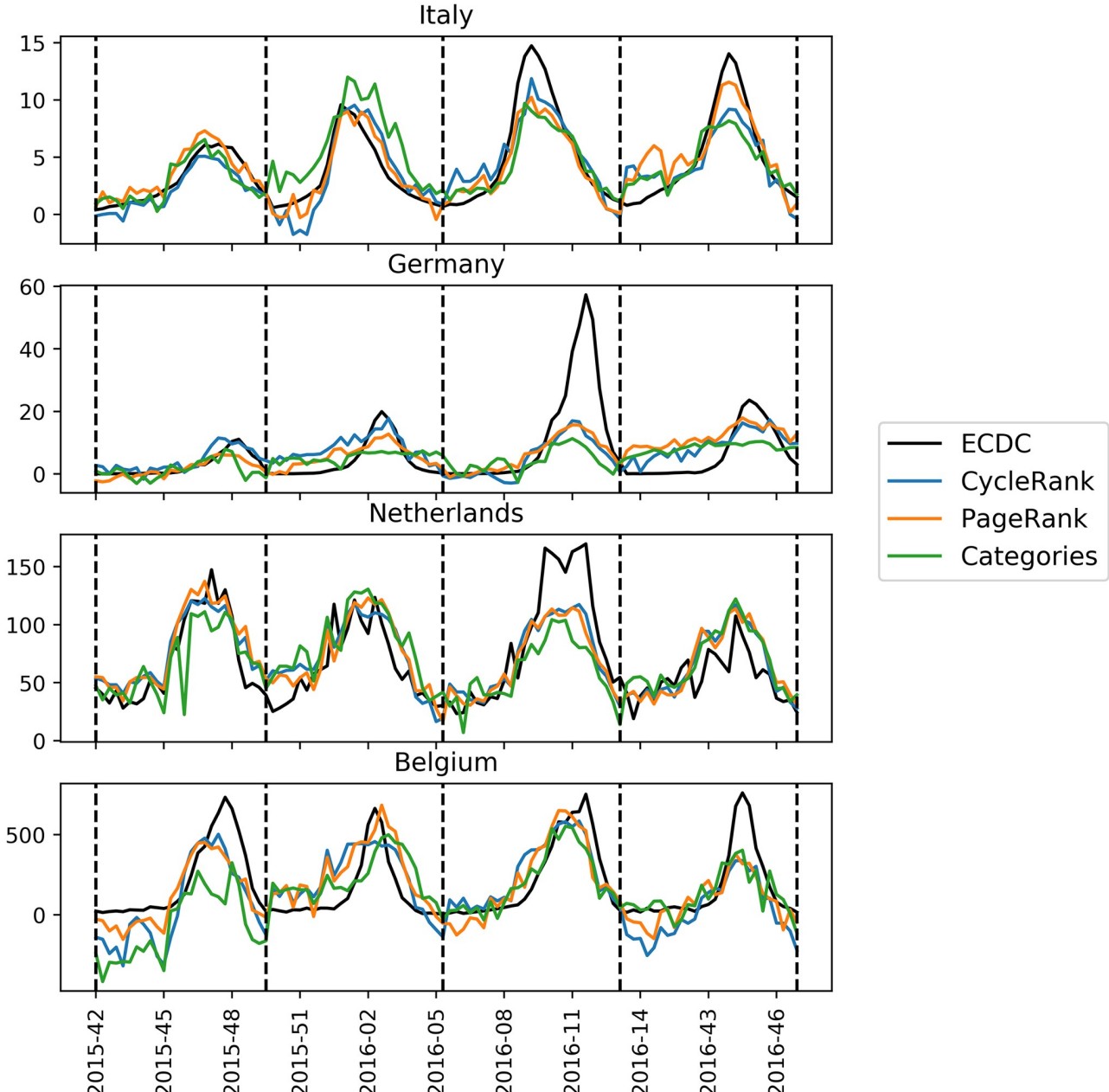

**Fig 1. Best *CycleRank*, *PPageRank* and *Categories* LASSO models predictions on the Italian, German, Belgian and Dutch influenza incidence.** The dashed lines delimit the various influenza seasons.

ILI incidence with Wikipedia? Are the selected predictors (Wikipedia pages) related in some way to the *"Influenza"* topic? Is there a difference between using hand-picked features or automatically discovered ones?

## 6.1 Top-5 Wikipedia's predictors

First, we focus on the estimated relative importance of each predictor. We wanted to find which pages are the most useful to estimate the ILI incidence of a given week. We focused on the weights assigned to the various Wikipedia pages by the models, LASSO and GLM. We

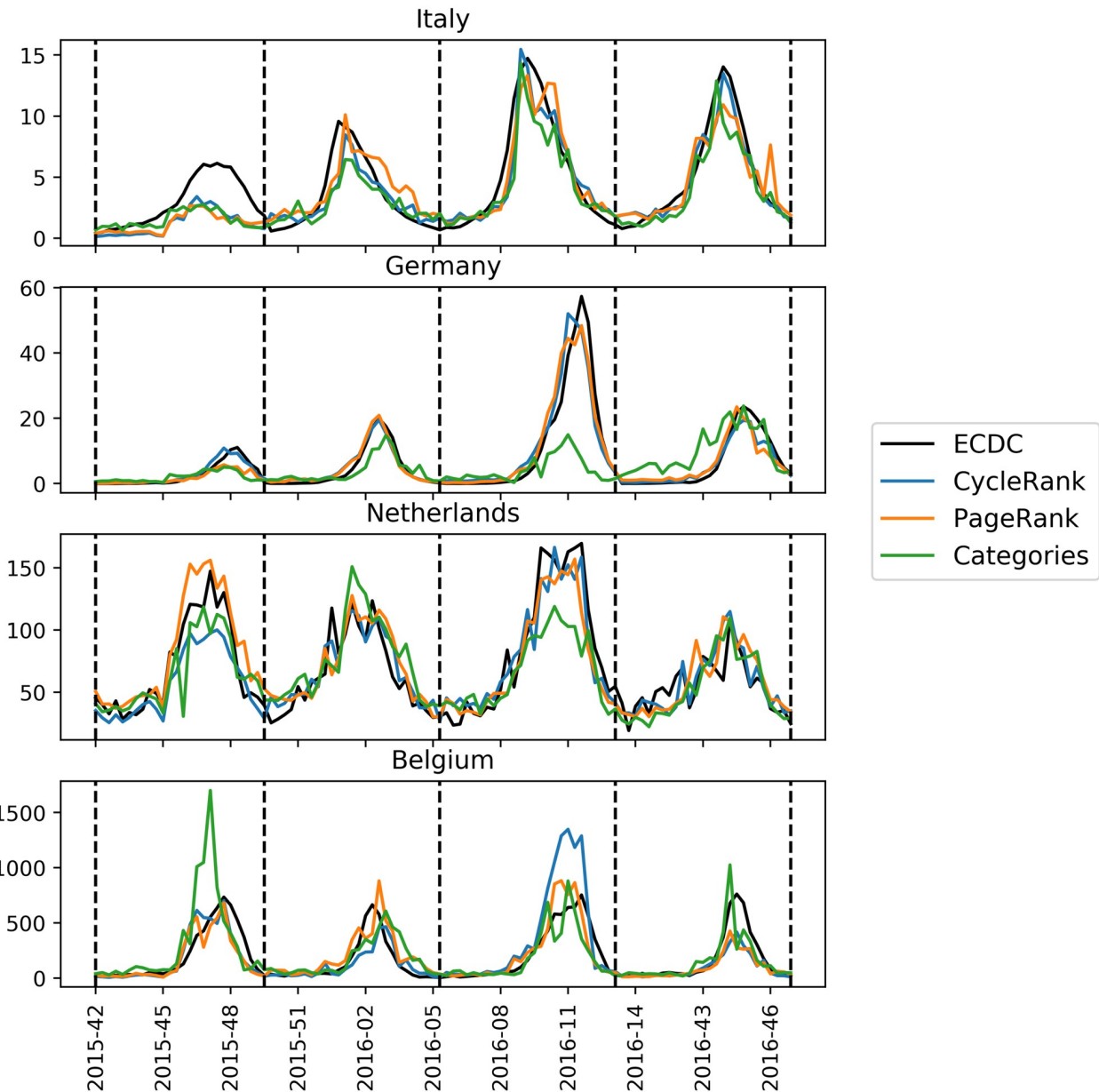

**Fig 2. Best *CycleRank*, *PPageRank* and *Categories* GLM models predictions on the Italian, German, Belgian and Dutch influenza incidence.** The dashed lines delimit the various influenza seasons.

directed our attention to those features which received a positive weight. A positive weight means that there is a positive correlation between the predictor and the influenza incidence.

It is also possible to analyze the effect of predictors with a negative weight. However, they present a challenge since it is harder to correlate them with real-world incidence. Generally, we argue that they usually compensate and balance the positive predictors as additional regularization terms.

To detect the most successful subset of features for the final model, we ranked the features based on their mean weight over the different models. If a given Wikipedia page is selected

**Table 6. Number of influenza peaks predicted correctly (#*peaks* = 4).** The bold values indicate the best results over all combinations of models and datasets. The first value in each cell indicates how many peaks the models predicted correctly. The second value indicates how many peaks were predicted in the ±2 weeks range. The green colour indicates the best results by either using the `PV` or the `PC+PV` dataset, by keeping both the country and the model fixed (no colour means no changes).

| Country | CycleRank | | PPageRank | | Categories | |
|---|---|---|---|---|---|---|
| | PV | PC+PV | PV | PC+PV | PV | PC+PV |
| IT | 1 (3) | 2 (3) | 0 (3) | **2 (4)** | 1 (1) | 1 (3) |
| DE | **1 (3)** | 0 (2) | 0 (2) | 0 (3) | 0 (2) | 0 (3) |
| NL | 1 (3) | 1 (3) | 0 (4) | **1 (4)** | 1 (1) | 1 (3) |
| BE | 0 (3) | **2 (4)** | 2 (3) | 0 (2) | 1 (3) | 0 (2) |
| Country | CycleRank-GLM | | PageRank-GLM | | Categories-GLM | |
| | PV | PV+PC | PV | PV+PC | PV | PV+PC |
| IT | 1 (3) | 1 (3) | 2 (3) | 0 (3) | 0 (4) | 2 (3) |
| DE | 1 (4) | 2 (4) | 1 (4) | 2 (4) | 1 (3) | 1 (4) |
| NL | 1 (2) | 2 (0) | 1 (3) | 1 (3) | 1 (3) | 1 (2) |
| BE | 1 (4) | 2 (4) | 2 (3) | 1 (3) | 1 (1) | 4 (0) |

multiple times by separate models for different influenza seasons, we found a feature that can predict the ILI incidence over many seasons.

We collected the top-5 features found by each model type (*CycleRank*, *PPageRank* and *Categories*), both from LASSO and GLM. We also computed the shortest path distance $D_I$ between the *"Influenza"* page to the found predictor and the Pearson Correlation Coefficient (PCC) for each chosen page against the influenza incidence. Unfortunately, the predictors identified by the GLM models are difficult to interpret. Namely, the non-linearity applied by the link function makes it harder to extrapolate clear information from the predictors with the highest weight since the relationship between the predictors and the influenza incidence is now less direct. Therefore, we did not show any analysis on them. The complete results can be seen in Table 7. We can notice how the top-5 predictors of many models are Wikipedia pages related to the influenza topic from the results. If we perform a rapid semantic analysis, some of them refer directly to the symptoms (e.g., *"Fever"*, *"Bronchitis"*), the pathogens (e.g., *"Influenza A virus subtype H1N1"*) or to similar concepts related to ILI (e.g., *"Antiviral Drug"*, *"Swine Influenza"*). We can also identify some super-predictors. Namely, some features are chosen multiple times by several models (e.g., *"Febbre"* for the Italian models or *"Grippe"* for both the Belgian and Dutch models). These super-predictors tend to have a very high PCC. Thus, they are very valid in predicting the variation of the ILI incidence. Ultimately, 9 out the best 12 models considered in Table 7 show to have at least half of their top-5 predictors with a distance $D_I$ less or equal than 2 from the *"Influenza"* page. More interestingly, the *Categories* models show that they tend to find predictors further away from the *Influenza* page.

## 6.2 Size of the feature subsets

Table 8 shows the minimum, the maximum and the mean number of features selected by each model during training. Generally, we can observe that the final models always produce subsets of the original feature sets. Not all the Wikipedia pages available are good predictors, and selecting too many features would cause a loss of generalization. Interestingly, the GLM models select very few features, and they tend to prefer smaller predictor sets than the LASSO models. This can be noticed by looking at the GLM's Italian and German data. This can also be an

**Table 7. Top-5 Wikipedia pages selected by the LASSO models for the influenza seasons from 2015 to 2019 for all the examined countries.** We report only the models which performed better (see Table 3 for the best models). For each model, we report the page name, the shortest-path distance $D_I$ between the page and the corresponding *"Influenza"* page, and the Pearson Correlation Coefficient (PCC) measured against the influenza incidence. We also report the corresponding page in the English Wikipedia in parentheses. We used the value NE to specify when a page has no English equivalent. The value $D_I > 3$ indicates that the page is more than three hops away from the *"Influenza"* page.

**(a)** Italy

| Cyclerank | | | PPageRank | | | Categories | | |
|---|---|---|---|---|---|---|---|---|
| Page Name | PCC | $D_I$ | Page Name | PCC | $D_I$ | Page Name | PCC | $D_I$ |
| Febbre (Fever) | 0.55 | 1 | Febbre (Fever) | 0.55 | 1 | Febbre (Fever) | 0.55 | 1 |
| Bronchite (Bronchitis) | 0.39 | 1 | Polmonite (Pneumonia) | 0.45 | 1 | Influenza asiatica (1957–1958 influenza pandemic) | 0.67 | >3 |
| Neoplasia (Neoplasm) | 0.36 | 1 | Antibiotico (Antibiotic) | 0.50 | 1 | Influenza di Hong Kong (Hong Kong flu) | 0.70 | >3 |
| Tosse post-virale (Post-viral cough) | 0.78 | >3 | Bronchite (Bronchitis) | 0.39 | 1 | Influenza suina (Swine influenza) | 0.25 | >3 |
| Virus respiratorio sinciziale umano (Respiratory syncytial virus) | 0.60 | >3 | Neoplasia (Neoplasm) | 0.36 | 1 | Bronchite (Bronchitis) | 0.39 | 1 |

**(b)** Germany

| Cyclerank | | | PPageRank | | | Categories | | |
|---|---|---|---|---|---|---|---|---|
| Page Name | PCC | $D_I$ | Page Name | PCC | $D_I$ | Page Name | PCC | $D_I$ |
| Influenza-Schnelltest (Rapid influenza diagnostic test) | 0.94 | 1 | Influenza-Schnelltest (Rapid influenza diagnostic test) | 0.94 | 1 | Bradykardie (Bradycardia) | 0.66 | 2 |
| Pneumokokken (Streptococcus pneumoniae) | 0.63 | 1 | Superinfektion (Superinfection) | 0.65 | 1 | Impfstoff (Vaccine) | 0.09 | 2 |
| Superinfektion (Superinfection) | 0.65 | 1 | Asiatische Grippe (1957–1958 influenza pandemic) | 0.79 | >3 | Toxoidimpfstoff (Toxoid) | 0.27 | >3 |
| Virostatikum (Antiviral drug) | 0.40 | 2 | Influenza-A-Virus H1N1 (Influenza A virus subtype H1N1) | 0.68 | >3 | Virusinfektion (Viral disease) | 0.38 | 2 |
| Pferdeinfluenza (Equine influenza) | 0.31 | 2 | Rimantadin (Rimantadine) | 0.54 | 1 | Akute Bronchitis (Acute bronchitis) | 0.54 | >3 |

**(c)** Belgium

| Cyclerank | | | PPageRank | | | Categories | | |
|---|---|---|---|---|---|---|---|---|
| Page Name | PCC | $D_I$ | Page Name | PCC | $D_I$ | Page Name | PCC | $D_I$ |
| Griep (Influenza) | 0.59 | 1 | Griep (Influenza) | 0.59 | 1 | Flavivirus (Flaviviridae) | 0.33 | >3 |
| Koorts (Fever) | 0.45 | 1 | Lichaamstemperatuur (Body temperature) | 0.30 | 2 | Mycovirus (Mycovirus) | 0.16 | >3 |
| Immuunsysteem (Immune system) | 0.04 | 1 | Urine (Urine) | 0.12 | 2 | Triple gene block (NE) | 0.17 | >3 |
| Immuundeficiëntie (Immunodeficiency) | -0.12 | 1 | Inenting (Vaccination) | -0.08 | 2 | Parotitis (Parotitis) | -0.15 | >3 |
| Ziekte (Disease) | -0.05 | 2 | B-cel (B cell) | -0.01 | 2 | Epidermodysplasia verruciformis (Epidermodysplasia verruciformis) | 0.00 | >3 |

**(d)** Netherlands

| Cyclerank | | | PPageRank | | | Categories | | |
|---|---|---|---|---|---|---|---|---|
| Page Name | PCC | $D_I$ | Page Name | PCC | $D_I$ | Page Name | PCC | $D_I$ |
| Griep (Influenza) | 0.75 | 1 | Griep (Influenza) | 0.75 | 1 | Griep (Influenza) | 0.75 | 1 |
| Hongkonggriep (Hong Kong flu) | 0.51 | 1 | Immuniteit (NE) | 0.14 | >3 | Aviair reovirus (Avian reovirus) | -0.04 | >3 |
| Aziatische griep (1957–1958 influenza pandemic) | 0.61 | >3 | Hongkonggriep (Hong Kong flu) | 0.51 | 1 | Geleidingsafasie (Conduction aphasia) | -0.01 | >3 |
| Opportunistische infectie (Opportunistic infection) | -0.07 | >3 | 1970-1979 (1970s) | -0.04 | 2 | Krim-Congo-hemorragische koorts (Crimean–Congo hemorrhagic fever) | 0.45 | >3 |
| Varkensgriep (Swine influenza) | 0.21 | 1 | Basalecelcarcinoom (Basal-cell carcinoma) | -0.19 | >3 | Hepatitis B (Hepatitis B) | 0.17 | >3 |

effect of the different penalties used since GLM models employ elastic net. The GLM's Belgian and Dutch models show similar behaviour, although they have a mean size closer to their LASSO counterparts. In conclusion, we argue that we need to monitor just a few pages to be able to obtain reasonable estimates of the ILI incidence.

**Table 8. Analysis of the number of features selected by the LASSO and GLM models for each influenza season.** We record the minimum, the maximum and the mean number of features for all the countries and the different datasets used.

| Country | CycleRank | | | | | | PageRank | | | | | | Categories | | | | | |
|---|---|---|---|---|---|---|---|---|---|---|---|---|---|---|---|---|---|---|
| | PV | | | PV+PC | | | PV | | | PV+PC | | | PV | | | PV+PC | | |
| | Min | Max | Mean | Min | Max | Mean | Min | Max | Mean | Min | Max | Mean | Min | Max | Mean | Min | Max | Mean |
| IT | 28 | 65 | 48.25 | 49 | 100 | 72.25 | 26 | 53 | 36.00 | 91 | 136 | 109.00 | 19 | 81 | 50.75 | 26 | 67 | 43.25 |
| DE | 6 | 63 | 32.25 | 35 | 62 | 46.25 | 25 | 68 | 42.50 | 13 | 39 | 20.50 | 34 | 61 | 44.00 | 5 | 69 | 33.25 |
| NL | 22 | 43 | 30.75 | 9 | 17 | 12.50 | 18 | 46 | 27.50 | 22 | 47 | 30.25 | 21 | 60 | 38.25 | 45 | 107 | 69.75 |
| BE | 3 | 30 | 16.75 | 16 | 48 | 29.50 | 10 | 47 | 23.50 | 23 | 60 | 45.00 | 26 | 65 | 37.00 | 3 | 7 | 5.25 |
| Country | CycleRank-GLM | | | | | | PageRank-GLM | | | | | | Categories-GLM | | | | | |
| | PV | | | PV+PC | | | PV | | | PV+PC | | | PV | | | PV+PC | | |
| | Min | Max | Mean | Min | Max | Mean | Min | Max | Mean | Min | Max | Mean | Min | Max | Mean | Min | Max | Mean |
| IT | 1 | 6 | 4.75 | 4 | 6 | 5 | 4 | 5 | 4.75 | 4 | 6 | 5 | 4 | 6 | 4.50 | 4 | 5 | 4.75 |
| DE | 4 | 10 | 6.50 | 2 | 3 | 2.75 | 3 | 6 | 4.50 | 2 | 3 | 2.75 | 7 | 11 | 8.75 | 7 | 10 | 8.25 |
| NL | 12 | 16 | 14.25 | 11 | 15 | 12.75 | 22 | 24 | 23.50 | 19 | 21 | 20 | 25 | 32 | 28 | 23 | 32 | 28.25 |
| BE | 24 | 27 | 25.50 | 19 | 27 | 22.75 | 32 | 38 | 35.50 | 33 | 47 | 40.25 | 37 | 43 | 40.25 | 34 | 41 | 36.25 |

## 6.3 Features shared between models

We were also interested in investigating internal predictor variations which may occur when training models by changing the initial feature set. Figs 3 and 4 show how many features are shared by the final trained GLM and LASSO models. First of all, if we compare the models trained with both the available datasets, PV and PC+PV, but with a fixed feature set (e.g., using only *CycleRank*), we can notice how they will both select very close features. For instance, for the LASSO Italian models, the *CycleRank* and *PPageRank* will share up to 93% features between using PC+PV or PV. We can notice this behaviour also for GLM models. This fact is interesting because it tells us that providing a better dataset with more influenza seasons, such as PC+PV, affects the final predictors, and it can push the models toward choosing different Wikipedia pages. These changes, in turn, could lead to an improvement in the performances.

In all the other cases, we can notice how the various models overlap just in some features. The models end up selecting almost non-intersecting sets of Wikipedia voices. For instance, in the GLM models, *Categories* share only up to 14% of the final predictors with *CycleRank* and *PPageRank*, regardless of the training dataset used. Despite this, we can retain comparable performances. Again, this behaviour suggests that there are just a few Wikipedia pages worth monitoring to estimate the ILI incidence.

## 7 Discussion

The methods outlined in the previous sections complement more traditional techniques, especially if these conventional approaches present a lag between collecting the data and the publications of the results. For instance, some illnesses require conducting some laboratory tests such to have a positive/negative response. It could be beneficial to employ these Wikipedia-based solutions to gain rapidly additional (albeit maybe less precise) information in such scenarios. More importantly, we have shown the feasibility of **using automatic methods to extract relevant predictors from Wikipedia itself by exploiting Wikipedia's links structure**. Wikipedia's structure is highly flexible, and its pages could be removed or changed very frequently, thus making some pages less relevant for the ILI estimation. Therefore, we may need to periodically retrain the models by incorporating new influenza data and additional Wikipedia pages to ensure proper stability. We have seen how just a small subset of pages is

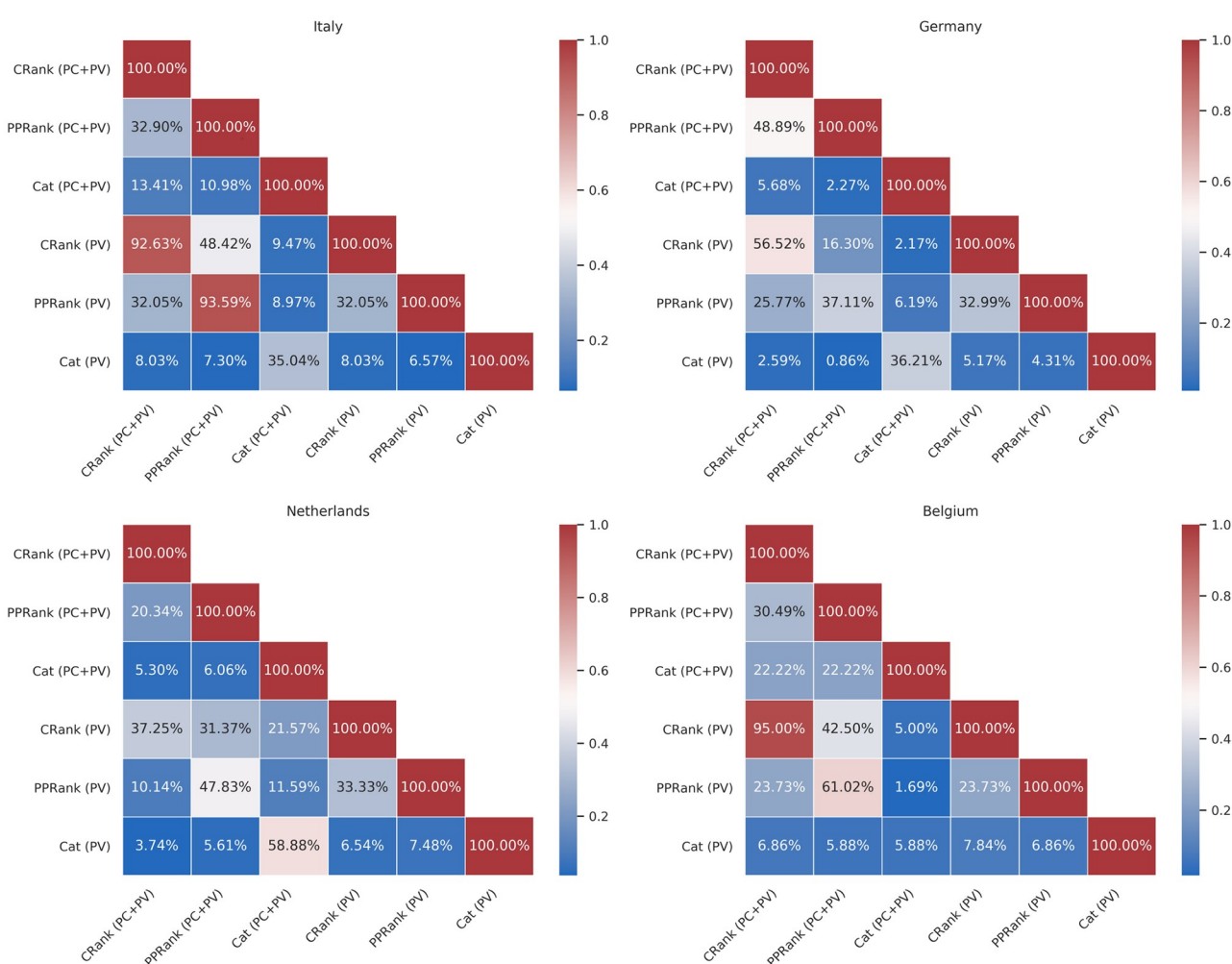

**Fig 3. Percentage of features shared by each LASSO model (CRank = *CycleRank*, PPRank = *PPageRank*, Cat = *Categories*), trained with the two different dataset (`PV` and `PC+PV`).** We recorded all the features selected by each model for each influenza season (a feature was included if its weight in at least one model was different from zero). Then, we computed the intersection between each set of features for each combination model/dataset.

responsible for a correct ILI estimation. Therefore we need to be able to find them correctly. *CycleRank* and *PPageRank* allow us to automatically select new subsets of pages, thus reducing the chances of having a low-performing model. Besides, we can employ these methods to effortlessly deploy machine learning models in other languages, thus making it easier to predict ILI incidence in different countries, as long as there is a Wikipedia's version with the same target language. Previously, this would have required gathering a team of experts in the target language to create a new feature dataset. Ultimately, we argue that this procedure is suitable for other illnesses, but further studies and analysis are required.

We discuss here some of the main shortcomings of our proposed approach that we leave to future work. Firstly, we employed simple linear regression as a baseline since it is easier to train, and it provides interpretable results. However, by using linear models, we are implying that we can obtain the response variable, the ILI incidence, from the linear combination of the predictors. This assumption may not hold very well in practice. We partly addressed this obstacle by showing how using a Poisson Generalized Linear Model can mitigate these issues and provide better performance. Secondly, one of the main issues of Wikipedia-based

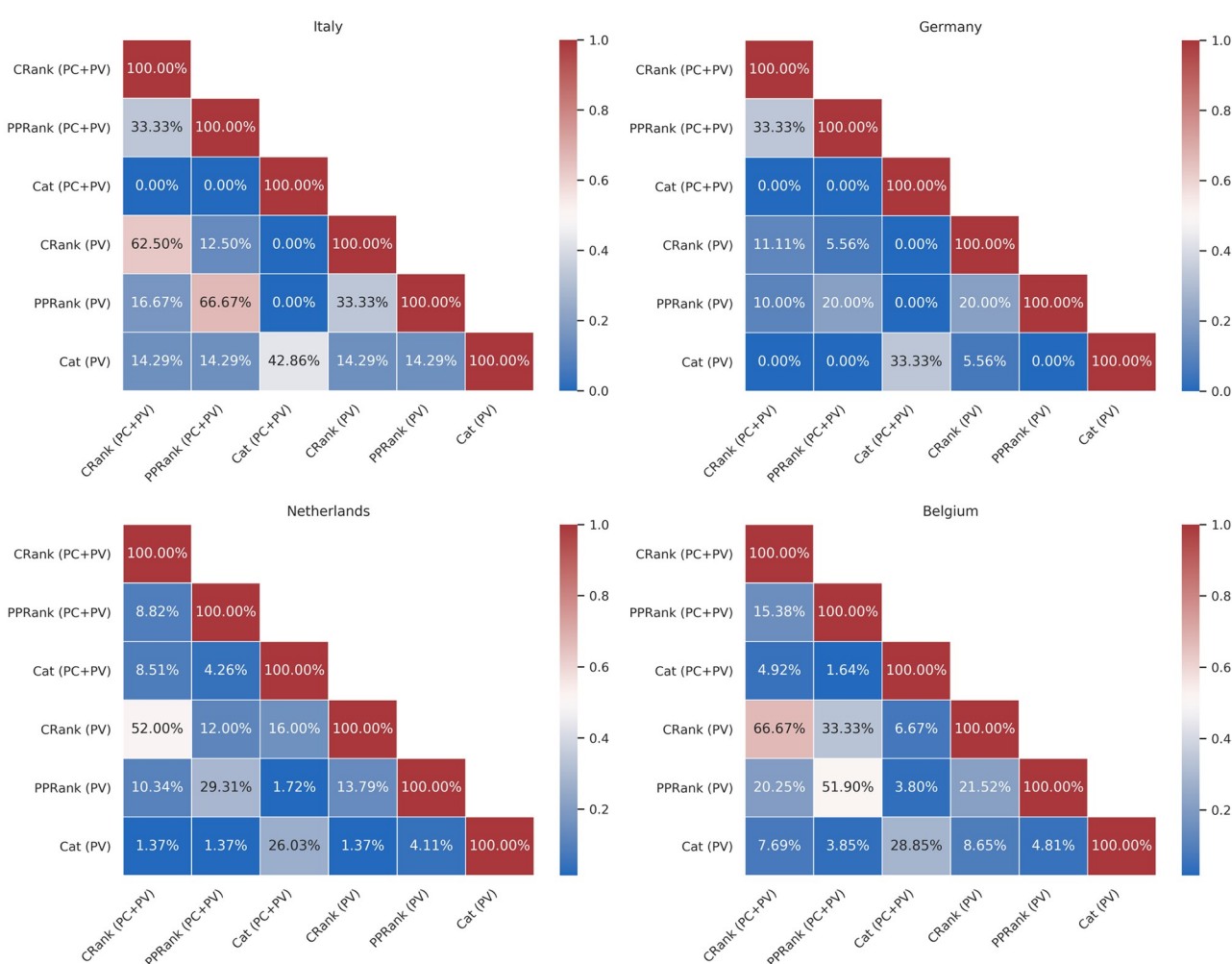

**Fig 4. Percentage of features shared by each GLM model (CRank = *CycleRank*, PPRank = *PPageRank*, Cat = *Categories*), trained with the two different dataset (`PV` and `PC+PV`).** We recorded all the features selected by each model for each influenza season (a feature was included if and only its weight in at least one model was different than zero). Then, we computed the intersection between each set of features for each combination model/dataset.

models is that they are sensitive to the media coverage of the target topics. This excessive coverage could lead to the overestimation of flu incidence because of the rising public attention. In our opinion, this is one of the primary concerns and limits of this class of methods which relies on human-generated web traffic. This issue was also the reason behind the failure of several other similar solutions, like Google Flu Trends [50, 51]. We can account for these concerns by extending our model to take into consideration the media coverage for the *"Influenza"* topic. We could perform anomaly detection on the page views, and we could instruct the model to normalize the various features to compensate for increased traffic. However, it is not trivial to monitor media and the news. We would need to identify reliable sources, and they should provide an open dataset as Wikipedia does. Moreover, we may need to employ natural language processing analysis to check if the news subject is semantically related to ILI. Lastly, another open challenge is understanding how data-driven models, such as LASSO or GLM, compare against classical compartmental models, such as SIR, SIRD or SEIR. Although we have shown how leveraging big data can deliver exciting results, we

did not delve into a proper comparison between the two techniques. This will require further tests and investigation.

Finally, we discuss some practical considerations in the case of production-ready systems. It might be advisable to develop an ensemble of multiple models that can leverage different data sources to improve the quality of the predictions. For example, Google Trends (https://trends.google.com/trends/) data could still complement and augment Wikipedia page views. However, in this work, we focused only on the latter for two main reasons: by using Wikipedia data, we can better control and understand the features we are dealing with. Google provides only the normalized trend, but we cannot control any intermediate processing or collection process. Lastly, by incorporating Google data, we might defeat the entire purpose of understanding the extent of Wikipedia's ability to predict the influenza trend.

## 8 Conclusion

The recent pandemics have shown the importance of estimating the spread of diseases in a fast and reliable way. In our case, we focused on the estimation of Influenza-Like Illnesses in European countries by using machine learning models and the page views coming from Wikipedia, the online encyclopedia. We extended upon state of the art by tackling the problem of using different language editions of Wikipedia visited by users in other European countries. Wikipedia is the first stop for many health-related searches. When people are sick and look for information on the internet, they are likely to read Wikipedia. In this work, we have shown that we can exploit this information to build linear and generalized linear models to do nowcasting of the incidence of ILI in a given country. The incidence estimation concerns both the total number of cases and influenza peak detection, namely, the week in which we will see the highest number of infected people in all the influenza seasons. We can use these results to direct the public authorities' possible efforts and devise safety measures in concert with more traditional approaches (e.g., SIR models, collection of laboratory tests, etc.). We tested our method on the previous four influenza seasons (from 2015 to 2019). More importantly, as the main contribution, we have also shown how it is possible to use two automated methods, *CycleRank* and *PPageRank*, to extract automatically and rapidly highly relevant features from Wikipedia by looking at Wikipedia's links structure. In previous works, this time-consuming task relied upon a group of expert physicians who manually chose every relevant Wikipedia page. The main advantage of *CycleRank* and *PPageRank* methods depends on their ability to generalize over multiple languages (and countries) without the need for expert supervision. We compared the performance of these novel methods against a more traditional one, called *Categories*, which uses hand-picked pages. We showed that the models trained with these newly found features are equal or even outperform the models trained with hand-picked Wikipedia's pages.

## Acknowledgments

The authors would like to thank Michele Bortolotti and the team of "Gestione Sistemi" at the University of Trento for their support with the HPC cluster.

## Author Contributions

**Conceptualization:** Giovanni De Toni, Cristian Consonni, Alberto Montresor.

**Data curation:** Giovanni De Toni, Cristian Consonni.

**Investigation:** Giovanni De Toni.

**Resources:** Alberto Montresor.

**Supervision:** Cristian Consonni, Alberto Montresor.

**Validation:** Giovanni De Toni.

**Visualization:** Giovanni De Toni.

**Writing – original draft:** Giovanni De Toni, Cristian Consonni, Alberto Montresor.

**Writing – review & editing:** Giovanni De Toni, Cristian Consonni, Alberto Montresor.

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
