## [Decision Letter · Decision Letter 0]

22 Jan 2021

PONE-D-20-37326

A general method for estimating the prevalence of Influenza-Like-Symptoms with Wikipedia data

PLOS ONE

Dear Dr. De Toni,

Thank you for submitting your manuscript to PLOS ONE. After careful consideration, we feel that it has merit but does not fully meet PLOS ONE’s publication criteria as it currently stands. Therefore, we invite you to submit a revised version of the manuscript that addresses the points raised during the review process.

Both reviewers raised a number of important issues that must be addressed point by point. I want to add that the details of the training and validation of the machine learning methods are lacking. For instance, no details whatsoever are given as to how crossvalidation was pursued. Also, in choosing a linear regression, why do you assume that "the correlation between the influenza incidence and the page views of Wikipedia was linear"? Do you have supporting evidence for that assumption? At least provide a little bit of an intuition for that choice. One may think that the correlation is not linear, for instance that there would be more of threshold behavior, whereby only after a certain flu incidence would page views correlate, no?

We look forward to receiving your revised manuscript.

Kind regards,

Luis M. Rocha, Ph.D.

Academic Editor

PLOS ONE

Journal Requirements:

Reviewers' comments:

Reviewer's Responses to Questions

**Comments to the Author**

1. Is the manuscript technically sound, and do the data support the conclusions?

Reviewer #1: Partly

Reviewer #2: Partly

2. Has the statistical analysis been performed appropriately and rigorously? 

Reviewer #1: No

Reviewer #2: Yes

3. Have the authors made all data underlying the findings in their manuscript fully available?

Reviewer #1: Yes

Reviewer #2: Yes

4. Is the manuscript presented in an intelligible fashion and written in standard English?

Reviewer #1: Yes

Reviewer #2: Yes

5. Review Comments to the Author

Reviewer #1: The present manuscript presents a methodology to nowcast influenza based on Wikipedia page visits. The method is interesting, the subject timely, and has the potential to be broadly applied to different languages. Using human curation to pre-select features is also a potentially relevant approach. However, the manuscript as presented has some concerning issues, discussed below. Moreover, the authors' bold claim in the discussion that they "have shown the feasibility of using automatic methods to extract relevant predictors ..." has no support. None of the feature-selection methods are new and the model performance is poor. Below is a point by point discussion of the paper.

Major points

1. Nowcasting: The model is intended to perform nowcasting, which typically implies training ans testing models on past data to monitor ongoing outbreaks. However, the authors are testing the model against whole seasons where the data is standardized (at least the Wikipedia dataset is - line 82). How could this be achieved without having the full data of the season? How well would the model(s) perform in a scenario of "true" nowcasting (ie, predict the next week(s) based on the last few weeks)?

2. Quality of the models: The prediction of the peak week is more interesting from the point of view of nowcasting (although concerns about standardization remain). However, judging it in relation to the ground truth is only one of the relevant metrics. Nowadays medical systems make predictions about peaks based on epidemiological data and models. How better is your model in relation to these more traditional approaches?

Unfortunately the models behave very poorly against the ground truth (table 4). This raises the question of how useful are they. Also, the Categories model misses a peak altogether in Germany (2nd panel of fig.1) and this should be discussed.

3. Differences between tools: The interpretation of the differences between the two datasets PV and PC+PV is not straightforward. The authors discuss it in terms of having more or less data, concluding that, for cycle rank and categories, more data is better while for pagerank it is worse. There is not enough evidence to say this because PC+PV not only has more data but it has different data (ex: PC includes bot visits). Also, there is no obviously better method or dataset, with them methods working differently in different countries. This raises concerns regarding generabillity.

4. Comparison to previous works: there have been many other efforts of nowcasting using Google searches, Wikipedia visits, on-call phone systems, self-reported symptom apps, etc. The authors should discuss their findings and model in comparison to previous published work and how their approach improves current thinking.

Minor points

1. Page 2, starting at line 66 - It would be good to include some of Wikipedia's methodology to arrive at these numbers since provenance of visits is important for the results of the paper.

2. Why was performance measured only with mean Pearson correlation? I would like to see both R^2 and Mean squared error as well.

3. In section 6, feature analysis. It is unclear why the authors focus only on pages positively correlated with cases. Negative correlations can be just as useful (albeit more difficult to interpret).

The paper has multiple typos. Here is a non-exhaustive list:

Legend of table 1 should be "... German and Dutch translations..."

lijne 121 "...in order to avoid making unnecessary..."

line 129 till the end of that paragraph. Replace Wikipedia's pages with Wikipedia pages line 160/161 "...circular walks that start and end..."

line 348 "...require conducting..."

Finally, the quality of the figures is very poor but we assume it will be improved before publication.

Reviewer #2: This paper describes a work aimed at applying a machine learning model on Wikipedia’s page views of a selected group of articles to obtain accurate estimates of influenza-like illnesses incidence in four European countries: Italy, Germany, Belgium, and the Netherlands.

This effort falls in a decade long research line aimed at using non traditional data sources generated by digital platforms to track and now-cast the Influenza-like Illness circulation among the general population. The goal is to extend the already existing studies in the USA context to a broader Europe-focussed one.

It is a very interesting attempt that, on the other hand, has still some issues. Some comments are reported in the following:

- in the introduction, only a little of the decade-long efforts and scientific studies to nowcast and forecast influenza through digital data (not necessarily with machine learning) are mentioned.

- how do they avoid over fitting and using features that are highly correlated? Feature selection is performed by the machine learning approach but still for each model tens of features are used. Did the authors check for collinearity?

- can this approach be used for other countries/languages? Wikipedia provides the country from which the pageviews are generated, so even for pages in English or French, it is possible to disentangle the provenance of the clicks.

- what is the use in public health, given the not-so-accurate estimate of the peak? Would a different regression

model help on this?

- would the integration of these data with other digital data or traditional surveillance data help in

making a more accurate estimate of the peak?

See paper by Bronwsntein https://journals.plos.org/ploscompbiol/article?id=10.1371/journal.pcbi.1004513

and Perrotta: https://dl.acm.org/doi/10.1145/3038912.3052670

6. PLOS authors have the option to publish the peer review history of their article (what does this mean?). If published, this will include your full peer review and any attached files.

Reviewer #1: No

Reviewer #2: No

---

## [Author Response · Author response to Decision Letter 0]

25 Mar 2021

Response letter for paper # PONE-D-20-37326

A general method for estimating the prevalence of Influenza-Like-Symptoms with Wikipedia data

The authors would like to thank the anonymous reviewers for their time and effort to provide extensive reviews and invaluable feedback. This document lists the changes that we have made in response to the reviewers’ comments.

In the paper, the major changes have been highlighted in yellow/strikethrough. We also have improved the text and corrected typos in several places, as suggested, without highlighting.

Giovanni De Toni

Cristian Consonni

Alberto Montresor

Reviewer #1: 

The present manuscript presents a methodology to nowcast influenza based on Wikipedia page visits. The method is interesting, the subject timely, and has the potential to be broadly applied to different languages. Using human curation to pre-select features is also a potentially relevant approach. However, the manuscript as presented has some concerning issues, discussed below. Moreover, the authors' bold claim in the discussion that they

"have shown the feasibility of using automatic methods to extract relevant predictors ..." has no support. None of the feature-selection methods are new and the model performance is poor. Below is a point by point discussion of the paper.

Major points

R1.1 Nowcasting: The model is intended to perform nowcasting, which typically implies training and testing models on past data to monitor ongoing outbreaks. However, the authors are testing the model against whole seasons where the data is standardized (at least the Wikipedia dataset is - line 82). How could this be achieved without having the full data of the season? How well would the model(s) perform in a scenario of "true" nowcasting (ie, predict the next week(s) based on the last few weeks)?

A1.1 The standardization is performed separately for the train and test set. Namely, we compute the mean and standard deviation of the train set, and we use them to standardize singularly each week of the test set. Obviously, here we are assuming that our train and test sets are coming from a similar underlying generation process. Therefore, there is no information shared between the train and test sets. 

Our models perform nowcasting given the pageviews computed for the target week without taking into account the previous ones. This was a methodological choice, since we wanted to try to predict influenza levels based solely on the page views of a given week, without using any additional information. 

We updated the corresponding sections in the paper to better explain these points. 

R1.2 Quality of the models: The prediction of the peak week is more interesting from the point of view of nowcasting (although concerns about standardization remain). However, judging it in relation to the ground truth is only one of the relevant metrics. Nowadays medical systems make predictions about peaks based on epidemiological data and models. How better is your model in relation to these more traditional approaches?

Unfortunately the models behave very poorly against the ground truth (table 4). This raises the question of how useful are they. Also, the Categories model misses a peak altogether in Germany (2nd panel of fig.1) and this should be discussed.

A1.2 To address this issue, we have added to our work an additional generalized linear model taken from other relevant papers (e.g., McIver et al., 2014) which should serve as an additional comparison against the simple Lasso linear model.

R1.3 Differences between tools: The interpretation of the differences between the two datasets PV and PC+PV is not straightforward. The authors discuss it in terms of having more or less data, concluding that, for cycle rank and categories, more data is better while for pagerank it is worse. There is not enough evidence to say this because PC+PV not only has more data but it has different data (ex: PC includes bot visits). Also, there is no obviously better method or dataset, with them methods working differently in different countries. This raises concerns regarding generabillity.

A1.3 We have updated the sections in the paper to address this concern by adding new experiments and by using more sophisticated preprocessing techniques on the PV and PV+PC dataset. See Table 3 and Table 4 for the updated evaluation. The new results show with more clarity that there are indeed benefits when using more data to train the models (even if they are collected from a different source). 

R1.4 Comparison to previous works: there have been many other efforts of nowcasting using Google searches, Wikipedia visits, on-call phone systems, self-reported symptom apps, etc. The authors should discuss their findings and model in comparison to previous published work and how their approach improves current thinking.

A1.4 Other recent and state-of-the-art approaches generally come without disclosing the code used to train their models or without the dataset employed. For example, Google searches or other corporate data are not generally available to external researchers or the public. Therefore, this limits our ability to produce a fair comparison between our methods and the ones presented in the relevant literature. Moreover, even if the techniques are detailed precisely, the software used is usually proprietary, thus creating a challenge from a reproducibility standpoint. To at least provide some additional context, we updated the introduction of this work by adding some recent works to address also current methodologies and state-of-the-art methods.

R1.5 Minor points Page 2, starting at line 66 - It would be good to include some of Wikipedia's methodology to arrive at these numbers since provenance of visits is important for the results of the paper.

A1.5 The section 2 was updated to add an intuition about how the provenance counts are computed. For a more complete overview, we refer to the original source, Wikistats, for the complete description of the methodology used (https://stats.wikimedia.org/wikimedia/squids/SquidReportPageViewsPerLanguageBreakdown.htm).

R1.6 Why was performance measured only with mean Pearson correlation? I would like to see both R^2 and Mean squared error as well.

A1.6 In the revised version of our paper, we added the R^2 score. We did not report the MSE since the various incidence datasets had different distributions and numerosity, thus the relative MSE change between the various models could have been misleading. Therefore, we deem the MSE to add more confusion to the overall picture rather than additional insights. 

R1.7 In section 6, feature analysis. It is unclear why the authors focus only on pages positively correlated with cases. Negative correlations can be just as useful (albeit more difficult to interpret).

A1.7 We updated section 6.1 to address this concern. We argue that negative weights can be interpreted as a balancing factor for the positive weights assigned to certain features such as to provide better estimates which minimize the model’s MSE. 

R1.8 The paper has multiple typos. Here is a non-exhaustive list:

Legend of table 1 should be "... German and Dutch translations..."

line 121 "...in order to avoid making unnecessary..."

line 129 till the end of that paragraph. Replace Wikipedia's pages with Wikipedia pages line 160/161 "...circular walks that start and end..."

line 348 "...require conducting..."

A1.8 We have corrected the typos listed here and we have performed a detailed proofreading of the paper.

R1.9 Finally, the quality of the figures is very poor but we assume it will be improved before publication.

A1.9 We inserted in the revised manuscript a new version of the pictures. Their quality should be better now. 

Reviewer #2: 

This paper describes a work aimed at applying a machine learning model on Wikipedia’s page views of a selected group of articles to obtain accurate estimates of influenza-like illnesses incidence in four European countries: Italy, Germany, Belgium, and the Netherlands. This effort falls in a decade-long research line aimed at using non-traditional data sources generated by digital platforms to track and now-cast the Influenza-like Illness circulation among the general population. The goal is to extend the already existing studies in the USA context to a broader Europe-focussed one. It is a very interesting attempt that, on the other hand, has still some issues. Some comments are reported in the following:

R2.1 In the introduction, only a little of the decade-long efforts and scientific studies to nowcast and forecast influenza through digital data (not necessarily with machine learning) are mentioned.

A2.1 We updated the introduction of this work by adding some recent works to address also current methodologies and state-of-the-art methods.

R2.2 How do they avoid overfitting and using features that are highly correlated? Feature selection is performed by the machine learning approach but still for each model tens of features are used. Did the authors check for collinearity?

A2.2 Regularization techniques are used to ensure feature selection and to prevent overfitting. Namely, LASSO and Elastic-Net penalties push the models to give positive weights to pages which are highly correlated with the influenza incidence. Moreover, the penalties select a set of features which present little collinearity, since we want to maximize the amount of information given by each of them (e.g., if two features are correlated, then the penalties will push the model towards choosing just one of them, since they carry the same information). 

R2.3 Can this approach be used for other countries/languages? Wikipedia provides the country from which the pageviews are generated, so even for pages in English or French, it is possible to disentangle the provenance of the clicks.

A2.3 We argue that our approach is indeed multilingual and it can be effortlessly applied to other countries/languages. However, we would like to point out that the information about the provenance of visits to Wikipedia articles is not publicly provided, since it is highly sensitive information that would put Wikipedia readers’ privacy at risk. To ensure the reproducibility of our work we used only publicly available data. In the case of languages for which the page views provenance is more diverse (e.g., English, Arabic), this could indeed lead to a decrease in accuracy of the models. However, we point out that McIver et Al. 2014 were able to use English pageviews to predict accurately USA incidence without the need for disentangling the provenance of page views.

R2.4 What is the use in public health, given the not-so-accurate estimate of the peak? Would a different regression model help on this?

A2.4 Despite the peak prediction not being completely satisfactory, we argue that being able to know the influenza incidence with a reasonable accuracy for a given week is still an asset. These estimates can already be provided without the need for any additional process to be put in place for doctors and health professional laboratory tests or any further action from the public. This includes also people who do not interact with their local health system at all. Moreover, these data could give an overview of the epidemiological situation of a country in a matter of seconds. As we noted in the conclusion of our work, the purpose of these methods is to complement more traditional human activities, and we underline that no matter how sophisticated our model can be or how much data we possess, there are still common pitfalls and limitations that should be taken into account. These risks have already been out in the past, for example (Lazer et al., 2014) tackled the limitations of Google Flu Trend project. Our aim with this paper is to provide a fully reproducible work, because the data, the methods and the code are available for the community without any restriction.

R2.5 Would the integration of these data with other digital data or traditional surveillance data help in making a more accurate estimate of the peak?

A2.5 Yes, see the paper by (Santillana et al., 2015) and (Perrotta et al., 2017).

References

McIver DJ, Brownstein JS (2014) Wikipedia Usage Estimates Prevalence of Influenza-Like Illness in the United States in Near Real-Time. PLOS Computational Biology 10(4): e1003581. https://doi.org/10.1371/journal.pcbi.1003581

Daniela Perrotta, Michele Tizzoni, and Daniela Paolotti. 2017. Using Participatory Web-based Surveillance Data to Improve Seasonal Influenza Forecasting in Italy. In Proceedings of the 26th International Conference on World Wide Web (WWW '17). International World Wide Web Conferences Steering Committee, Republic and Canton of Geneva, CHE, 303–310. DOI:https://doi.org/10.1145/3038912.3052670

Santillana M, Nguyen AT, Dredze M, Paul MJ, Nsoesie EO, et al. (2015) Combining Search, Social Media, and Traditional Data Sources to Improve Influenza Surveillance. PLOS Computational Biology 11(10): e1004513. https://doi.org/10.1371/journal.pcbi.1004513

---

## [Decision Letter · Decision Letter 1]

12 May 2021

PONE-D-20-37326R1

A general method for estimating the prevalence of Influenza-Like-Symptoms with Wikipedia data

PLOS ONE

Dear Dr. De Toni,

Thank you for submitting your manuscript to PLOS ONE. After careful consideration, we feel that it has merit but does not fully meet PLOS ONE’s publication criteria as it currently stands. Therefore, we invite you to submit a revised version of the manuscript that addresses the points raised during the review process.

Thank you for thoroughly addressing the reviewer comments. The paper is much improved by addressing all the concerns. In the next version, please address the few reviewer major points still left. I will then confirm those were addressed and will not need to send paper to additional review for publications. Specifically:

A1.2 A mention of whether using Wikipedia data is better than traditional compartment models

A 1.3 the discussion of PV versus PV+PC a discussion of whether using more data improves the models.

A 1.4 Specify what is meant by "Google searches not being available to researchers".

We look forward to receiving your revised manuscript.

Kind regards,

Luis M. Rocha, Ph.D.

Academic Editor

PLOS ONE

Journal Requirements:

Reviewers' comments:

Reviewer's Responses to Questions

**Comments to the Author**

1. If the authors have adequately addressed your comments raised in a previous round of review and you feel that this manuscript is now acceptable for publication, you may indicate that here to bypass the “Comments to the Author” section, enter your conflict of interest statement in the “Confidential to Editor” section, and submit your "Accept" recommendation.

Reviewer #1: (No Response)

2. Is the manuscript technically sound, and do the data support the conclusions?

Reviewer #1: Yes

3. Has the statistical analysis been performed appropriately and rigorously? 

Reviewer #1: Yes

4. Have the authors made all data underlying the findings in their manuscript fully available?

Reviewer #1: Yes

5. Is the manuscript presented in an intelligible fashion and written in standard English?

Reviewer #1: Yes

6. Review Comments to the Author

Reviewer #1: The authors did a thorough job in addressing the concerns and the manuscript is significantly improved as a result. Particularly the addition of the GLM model helps justify the bolder claims.

Major points:

A 1.1 Lines 106 to 121 now address this concern. The mean and standard deviation used for the data transformation now come from the training dataset and not the test set.

A 1.2 I am afraid we were not clear in our previous comments. The medical authorities in each country typically use compartmental models based on SIR, that rely on information from previous weeks (and/or seasons). The question is whether using Wikipedia data to predict influenza is better than these traditional models. We understand this may be beyond the scope of the paper but would like to see it mentioned in the discussion. The GLM model visibly improves predictions and was a good addition to the paper.

A 1.3 the discussion of PV versus PV+PC is now more complete but did not fully address our concerns. The fact that PV and PC data are different, limits the discussion of “more vs. better”. For example, how would the results look like if only PC data was used? This is no longer a key claim of the paper, but we would still argue that the authors have not consistently shown that using more data improves the models.

A 1.4 The discussion of previous work is now more complete. As a small note, we are not sure what the authors mean about Google searches not being available to researchers. It is true that the underlying method/algorithm is not known, but normalized data is: https://trends.google.com/trends/

Minor points:

A1.5 Solved

A 1.6 Solved

A 1.7 This could be circumvented and used to improve the model but it’s a minor point, just for the author’s consideration

A 1.8 We noticed some more typos (nothing like fresh eyes) and list them below hoping there are useful:

line 45, should be "Center for Disease Control and Prevention (CDC)", singular.

line 85, the word “only” is repeated "The Wikimedia Foundation, which runs Wikipedia's servers, only provides information about countries only at an aggregated level"

line 413 Italian and German should be capitalized

line 414 same for Belgian and Dutch

A 1.9 Solved

7. PLOS authors have the option to publish the peer review history of their article (what does this mean?). If published, this will include your full peer review and any attached files.

Reviewer #1: No

---

## [Author Response · Author response to Decision Letter 1]

27 May 2021

Response letter for paper # PONE-D-20-37326

A general method for estimating the prevalence of Influenza-Like-Symptoms with Wikipedia data

The authors would like to thank the anonymous reviewers for their time and effort to provide these additional reviews and helpful feedback. This document lists the changes that we have made in response to the reviewer’s comments.

In the paper, the major changes have been highlighted in yellow/strikethrough. We also have improved the text and corrected additional typos in several places, as suggested, without highlighting.

Giovanni De Toni

Cristian Consonni

Alberto Montresor

Reviewer #1: 

The authors did a thorough job in addressing the concerns and the manuscript is significantly improved as a result. Particularly the addition of the GLM model helps justify the bolder claims.

Major points:

R1.1 I am afraid we were not clear in our previous comments. The medical authorities in each country typically use compartmental models based on SIR, that rely on information from previous weeks (and/or seasons). The question is whether using Wikipedia data to predict influenza is better than these traditional models. We understand this may be beyond the scope of the paper but would like to see it mentioned in the discussion. The GLM model visibly improves predictions and was a good addition to the paper.

A1.1 It is true that we do not provide a direct comparison between compartmental methods and our data-driven approach. We updated the “Discussion” session of the paper to mention this issue as a possible future extension. Our methodology is intended to be an addition to these standard techniques to support and diversify the sources from which we try to predict how the influenza trend will evolve. 

R1.2 the discussion of PV versus PV+PC is now more complete but did not fully address our concerns. The fact that PV and PC data are different, limits the discussion of “more vs. better”. For example, what would the results look like if only PC data was used? This is no longer a key claim of the paper, but we would still argue that the authors have not consistently shown that using more data improves the models.

A1.2 We updated the experiments in the paper by showing also the Pearson Correlation Coefficients of models trained by using only the PC dataset. From the results, it can be seen that models trained with the PC data underperform with respect to models trained with PV or PV+PC. Moreover, from a practical perspective, the PC dataset has been discontinued by Wikipedia and models trained with it may fall prey to distribution shift (since the collection methodology changed) or other phenomena which could make them less precise in the future. 

R1.3 The discussion of previous work is now more complete. As a small note, we are not sure what the authors mean about Google searches not being available to researchers. It is true that the underlying method/algorithm is not known, but normalized data is: https://trends.google.com/trends/

A1.3 We thank the reviewer for having pointed this out. We would like to expand on this and explain the practical reason why we did not consider Google data for this work:

- Even if those data are available to the public, Google might change their availability or restrict access, so scientists could not build models that can last over the years. We are aware that Wikipedia is not exempt from these issues in principle - in the sense that the project may somehow die in the long term - but we still think it offers better guarantees in terms of openness.

- Using Wikipedia data, we can better control and understand the features we are dealing with. Google provides only the normalized trend, but we have no control over any intermediate processing and the collection process.

- By incorporating Google data, we might defeat the entire purpose of understanding the extent of Wikipedia's ability to predict the influenza trend.

On the other hand, in the case of production-ready systems, it might be advisable to develop an ensemble of multiple models that can leverage different data sources to improve the quality of the predictions.

---

## [Editor Report · Decision Letter 2]

2 Jun 2021

PONE-D-20-37326R2

A general method for estimating the prevalence of Influenza-Like-Symptoms with Wikipedia data

PLOS ONE

Dear Dr. De Toni,

Thank you for submitting your manuscript to PLOS ONE. After careful consideration, we feel that it has merit but does not fully meet PLOS ONE’s publication criteria as it currently stands. Therefore, we invite you to submit a revised version of the manuscript that addresses the points raised during the review process.

Thank you for addressing the final reviewer comments. However, your response to R.1.3 (Google trends data) did not make it in any form to the text of the article, and it should. Can you summarize your points and add them to discussion in paper? On that note, I understand your second and third point (normalized data and studying Wikipedia data separately), but your first point (that Google may stop making google trends data available) is too much of a stretch... Google trends has been available for a long time. While, of course, Google can change that  in future, its availability has been as stable as Wikipedia for several years and indeed scientists include GT data in many working models. So the paper is best served by acknowledging that GT data could be included, but you chose not to due to it being normalized and wanting to study Wikipedia independently.

Another issue is that the paper would really profit from a final careful proofreading, ideally by a professional editor. Using the latter is your choice, but please do a careful editing for improving English readability. For instance, a passage such as:

"The models trained with the PC dataset result to be worse than the models trained with either PV or PV+PC. Since the PC dataset showed to be inferior to the other two datasets, we did not perform additional analysis on the models trained with it."

Could be simplified to:

"Since models trained with the PC dataset perform worse than models trained with either PV or PV+PC, we did not perform additional analysis of the former."

So I encourage a careful reading/editing to increase readability.

We look forward to receiving your revised manuscript.

Kind regards,

Luis M. Rocha, Ph.D.

Academic Editor

PLOS ONE
---

## [Author Response · Author response to Decision Letter 2]

9 Jul 2021

Response letter for paper # PONE-D-20-37326

A general method for estimating the prevalence of Influenza-Like-Symptoms with Wikipedia data

We would like to thank the anonymous reviewer for their time and effort in providing these additional reviews and helpful feedback. As we indicated in the submission system, we would like these revisions to be public, as we have found the process to be well-conducted and the feedback to be very beneficial to the quality of the paper. This document lists the changes that we have made in response to the reviewer's comments.

We added the additional comments regarding Google Flu Trends in the Discussion section, and we also worked to improve the manuscript's readability.

In the version, we highlighted significant changes in blue/strikethrough. We have also improved the text and corrected additional typos in several places, as suggested, without highlighting.

Giovanni De Toni

Cristian Consonni

Alberto Montresor

---

## [Editor Report · Decision Letter 3]

18 Aug 2021

A general method for estimating the prevalence of Influenza-Like-Symptoms with Wikipedia data

PONE-D-20-37326R3

Dear Dr. De Toni,

We’re pleased to inform you that your manuscript has been judged scientifically suitable for publication and will be formally accepted for publication once it meets all outstanding technical requirements. 

Thank you for addressing the final issues (especially the GT discussion). I apologize for some delay accepting this latest version, but I was on a family vacation for the last 2 weeks.

Kind regards,

Luis M. Rocha, Ph.D.

Academic Editor

PLOS ONE
---

## [Editor Report · Acceptance letter]

20 Aug 2021

PONE-D-20-37326R3 

A general method for estimating the prevalence of Influenza-Like-Symptoms with Wikipedia data 

Dear Dr. De Toni:

I'm pleased to inform you that your manuscript has been deemed suitable for publication in PLOS ONE. Congratulations! Your manuscript is now with our production department. 

Kind regards, 

on behalf of

Dr. Luis M. Rocha 

Academic Editor

PLOS ONE